# Cold Fusion: Training Seq2Seq Models Together with Language Models

## Abstract

Sequence-to-sequence (Seq2Seq) models with attention have excelled at tasks which involve generating natural language sentences such as machine translation, image captioning and speech recognition. Performance has further been improved by leveraging unlabeled data, often in the form of a language model. In this work, we present the Cold Fusion method, which leverages a pre-trained language model **during training**, and show its effectiveness on the speech recognition task. We show that Seq2Seq models with Cold Fusion are able to better utilize language information enjoying i) faster convergence and better generalization, and ii) almost complete transfer to a new domain while using less than 10% of the labeled training data.

## 1 Introduction

Sequence-to-sequence (Seq2Seq) (Bahdanau et al., 2015) models have achieved state-of-the-art results on many natural language processing problems including automatic speech recognition (Bahdanau et al., 2016; Chan et al., 2015), neural machine translation (Wu et al., 2016), conversational modeling (Vinyals & Le, 2015) and many more. These models learn to generate a variable-length sequence of tokens (e.g. texts) from a variable-length sequence of input data (e.g. speech or the same texts in another language). With a sufficiently large labeled dataset, vanilla Seq2Seq can model sequential mapping well, but it is often augmented with a language model to further improve the fluency of the generated text.

Because language models can be trained from abundantly available unsupervised text corpora which can have as many as one billion tokens (Jozefowicz et al., 2016; Shazeer et al., 2017), leveraging the rich linguistic information of the label domain can considerably improve Seq2Seq's performance. A standard way to integrate language models is to linearly combine the score of the task-specific Seq2Seq model with that of an auxiliary langauge model to guide beam search (Chorowski & Jaitly, 2016; Sutskever et al., 2014a). Gulcehre et al. (2015) proposed an improved algorithm called *Deep Fusion* that learns to fuse the hidden states of the Seq2Seq decoder and a neural language model with a gating mechanism, after the two models are trained independently.

While this approach has been shown to improve performance over the baseline, it has a few limitations. First, because the Seq2Seq model is trained to output complete label sequences without a language model, its decoder learns an implicit language model from the training labels, taking up a significant portion of the decoder capacity to learn redundant information. Second, the residual language model baked into the Seq2Seq decoder is biased towards the training labels of the parallel corpus. For example, if a Seq2Seq model fully trained on legal documents is later fused with a medical language model, the decoder still has an inherent tendency to follow the linguistic structure found in legal text. Thus, in order to adapt to novel domains, Deep Fusion must first learn to discount the implicit knowledge of the language.

In this work, we introduce *Cold Fusion* to overcome both these limitations. Cold Fusion encourages the Seq2Seq decoder to learn to use the external language model during training. This means that Seq2Seq can naturally leverage potentially limitless unsupervised text data, making it particularly proficient at adapting to a new domain. The latter is especially important in practice as the domain from which the model is trained can be different from the real world use case for which it is deployed. In our experiments, Cold Fusion can almost completely transfer to a new domain for the speech

recognition task with 10 times less data. Additionally, the decoder only needs to learn task relevant information, and thus trains faster.

The paper is organized as follows: Section 2 outlines the background and related work. Section 3 presents the Cold Fusion method. Section 4 details experiments on the speech recognition task that demonstrate Cold Fusion's generalization and domain adaptation capabilities.

## 2 BACKGROUND AND RELATED WORK

### 2.1 SEQUENCE-TO-SEQUENCE MODELS

A basic Seq2Seq model comprises an encoder that maps an input sequence $\mathbf{x} = (x_1, \ldots, x_T)$ into an intermediate representation $\mathbf{h}$, and a decoder that in turn generates an output sequence $\mathbf{y} = (y_1, \ldots, y_K)$ from $\mathbf{h}$ (Sutskever et al., 2014b). The decoder can also attend to a certain part of the encoder states with an attention mechanism. The attention mechanism is called hybrid attention (Chorowski et al., 2015b), if it uses both the content and the previous context to compute the next context. It is soft if it computes the expectation over the encoder states (Bahdanau et al., 2015) as opposed to selecting a slice out of the encoder states.

For the automatic speech recognition (ASR) task, the Seq2Seq model is called an *acoustic model* (AM) and maps a sequence of spectrogram features extracted from a speech signal to characters.

### 2.2 INFERENCE AND LANGUAGE MODEL INTEGRATION

During inference, we aim to compute the most likely sequence $\hat{\mathbf{y}}$:

$$\hat{\mathbf{y}} = \underset{\mathbf{y}}{\operatorname{argmax}} \log p(\mathbf{y}|\mathbf{x}). \tag{1}$$

Here, $p(\mathbf{y}|\mathbf{x})$ is the probability that the task-specific Seq2Seq model assigns to sequence $\mathbf{y}$ given input sequence $\mathbf{x}$. The $\operatorname{argmax}$ operation is intractable in practice so we use a left-to-right beam search algorithm similar to the one presented in Sutskever et al. (2014a). We maintain a beam of $K$ partial hypothesis starting with the start symbol $\langle s \rangle$. At each time-step, the beam is extended by one additional character and only the top $K$ hypotheses are kept. Decoding continues until the stop symbol $\langle /s \rangle$ is emitted, at which point the hypothesis is added to the set of completed hypotheses.

A standard way to integrate the language model with the Seq2Seq decoder is to change the inference task to:

$$\hat{\mathbf{y}} = \underset{\mathbf{y}}{\operatorname{argmax}} \log p(\mathbf{y}|\mathbf{x}) + \lambda \log p_{\mathrm{LM}}(\mathbf{y}), \tag{2}$$

where $p_{\mathrm{LM}}(\mathbf{y})$ is the language model probability assigned to the label sequence $\mathbf{y}$. Chorowski & Jaitly (2016); Wu et al. (2016) describe several heuristics that can be used to improve this basic algorithm. We refer to all of these methods collectively as ***Shallow Fusion***, since $p_{\mathrm{LM}}$ is only used during inference.

Gulcehre et al. (2015) proposed ***Deep Fusion*** for machine translation that tightens the connection between the decoder and the language model by combining their states with a parametric gating:

$$g_t = \sigma(v^\top s_t^{\mathrm{LM}} + b) \tag{3a}$$

$$s_t^{\mathrm{DF}} = [s_t; g_t s_t^{\mathrm{LM}}] \tag{3b}$$

$$y_t = \operatorname{softmax}(\mathrm{DNN}(s_t^{\mathrm{DF}})), \tag{3c}$$

where $s_t$, $s_t^{\mathrm{LM}}$ and $s_t^{\mathrm{DF}}$ are the states of the task specific Seq2Seq model, language model and the overall deep fusion model. In (3c), DNN can be a deep neural network with any number of layers. $[a; b]$ is the concatenation of vectors $a$ and $b$.

In Deep Fusion, the Seq2Seq model and the language model are first trained independently and later combined as in Equation (3). The parameters $v$ and $b$ are trained on a small amount of data keeping the rest of the model fixed, and allow the gate to decide how important each of the models are for the current time step.

| Model | Prediction |
|---|---|
| Ground Truth | where's the sport in that greer snorts and leaps greer hits the dirt hard and rolls |
| Plain Seq2Seq | where is the sport **and** that **through snorks** and leaps **clear its** the dirt **card** and **rules** |
| Deep Fusion | where is the sport **and** that **there is north some beliefs through its** the dirt **card** and **rules** |
| Cold Fusion | where's the sport in that greer snorts and leaps greer hits the dirt hard and rolls |
| Cold Fusion (Fine-tuned) | where's the sport in that greer snorts and leaps greer hits the dirt hard and rolls |
| Ground Truth | jack sniffs the air and speaks in a low voice |
| Plain Seq2Seq | **jacksonice** the air and **speech** in a **logos** |
| Deep Fusion | **jacksonice** the air and **speech** in a **logos** |
| Cold Fusion | jack sniffs the air and speaks in a low voice |
| Cold Fusion (Fine-tuned) | jack sniffs the air and speaks in a low voice |
| Ground Truth | skipper leads her to the dance floor he hesitates looking deeply into her eyes |
| Plain Seq2Seq | **skip er leadure** to the dance floor he **is it takes** looking deeply into her eyes |
| Deep Fusion | **skip er leadure** to the dance floor he **has it takes** looking deeply into her eyes |
| Cold Fusion | skipper leads **you** to the dance floor he **has a tates** looking deeply into her eyes |
| Cold Fusion (Fine-tuned) | skipper leads her to the dance floor he hesitates looking deeply into her eyes |

Table 1: Some examples of predictions by the Deep Fusion and Cold Fusion models.

The biggest disadvantage with Deep Fusion is that the task-specific model is trained independently from the language model. This means that the Seq2Seq decoder needs to learn a language model from the training data labels, which can be rather parsimonious compared to the large text corpora available for language model training. So, the fused output layer of (3) should learn to overcome this bias in order to incorporate the new language information. This also means that a considerable portion of the decoder capacity is wasted.

### 2.3 SEMI-SUPERVISED LEARNING IN SEQ2SEQ MODELS

A few methods have been proposed for leveraging unlabeled text corpora in the target domain, for both better generalization and domain transfer.

Sennrich et al. (2015) proposed ***backtranslation*** as a way of using unlabeled data for machine translation. Backtranslation improves the BLEU score by increasing the parallel training corpus of the neural machine translation model by automatically translating the unlabeled target domain text. However, this technique does not apply well to other tasks where backtranslation is infeasible or of very low quality (like image captioning or speech recogntion).

Ramachandran et al. (2016) proposed warm starting the Seq2Seq model from language models trained on source and target domains separately. ***Unsupervised pre-training*** shows improvements in the BLEU scores. Ramachandran et al. (2016) also show that this improvement is from improved generalization, and not only better optimization. While this is a promising approach, the method is potentially difficult to leverage for the transfer task since training on the parallel corpus could end up effectively erasing the knowledge of the language models. Both back-translation and unsupervised pre-training are simple methods that require no change in the architecture.

## 3 COLD FUSION

Our proposed Cold Fusion method is largely motivated from the Deep Fusion idea but with some important differences. The biggest difference is that in Cold Fusion, the Seq2Seq model is trained from scratch together with a fixed pre-trained language model.

Because the Seq2Seq model is aware of the language model throughout training, it learns to use the language model for language specific information and capture only the relevant information conducive to mapping from the source to the target sequence. This disentanglement can increase the effective capacity of the model significantly. This effect is demonstrated empirically in Section 4 where Cold Fusion models perform well even with a very small decoder.

We also improve on some of the modeling choices of the fusion mechanism.

1. First, both the Seq2Seq hidden state $s_t$ and the language model hidden state $s_t^{\mathrm{LM}}$ can be used as inputs to the gate computation. The task-specific model's embedding contains information about the encoder states which allows the fused layer to decide its reliance on the language model in case of input uncertainty. For example, when the input speech is noisy or a token unseen by the Seq2Seq model is presented, the fusion mechanism learns to pay more attention to the language model.

2. Second, we employ fine-grained (FG) gating mechanism as introduced in Yang et al. (2016). By using a different gate value for each hidden node of the language model's state, we allow for greater flexibility in integrating the language model because the fusion algorithm can choose which aspects of the language model it needs to emphasize more at each time step.

3. Third, we replace the language model's hidden state with the language model probability. The distribution and dynamics of $s_t^{\mathrm{LM}}$ can vary considerably across different language models and data. As a concrete example, any fusion mechanism that uses the LM state is not invariant to the permutation of state hidden nodes. This limits the ability to generalize to new LMs. By projecting the token distribution onto a common embedding space, LMs that model novel uses of the language can still be integrated without state discrepancy issues. This also means that we can train with or swap on $n$-gram LMs during inference.

The *Cold Fusion* layer (Figure 1) works as follows:

$$h_t^{\mathrm{LM}} = \mathrm{DNN}(\ell_t^{\mathrm{LM}}) \tag{4a}$$

$$g_t = \sigma(W[s_t; h_t^{\mathrm{LM}}] + b) \tag{4b}$$

$$s_t^{\mathrm{CF}} = [s_t; g_t \circ h_t^{\mathrm{LM}}] \tag{4c}$$

$$r_t^{\mathrm{CF}} = \mathrm{DNN}(s_t^{\mathrm{CF}}) \tag{4d}$$

$$\hat{P}(y_t|x, y_{<t}) = \mathrm{softmax}(r_t^{\mathrm{CF}}) \tag{4e}$$

$\ell_t^{\mathrm{LM}}$ is the logit output of the language model, $s_t$ is the state of the task specific model, and $s_t^{\mathrm{CF}}$ is the final fused state used to generate the output. Since logits can have arbitrary offsets, the maximum value is subtracted off before feeding into the layer. In (4a), (4d), the DNN can be a deep neural network with any number of layers. In our experiments, we found a single affine layer, with ReLU activation prior to softmax, to be helpful.

## 4 EXPERIMENTS

### 4.1 SETUP

For our experiments, we tested the Cold Fusion method on the speech recognition task. The results are compared using the character error rate (CER) and word error rate (WER) on the evaluation sets. For all models which were trained on the source domain, the source CER and WER indicate in-domain performance and the target CER and WER indicate out-of-domain performance.

We collected two data sets: one based on search queries which served as our *source* domain, and another based on movie transcripts which served as our *target* domain. For each dataset, we used

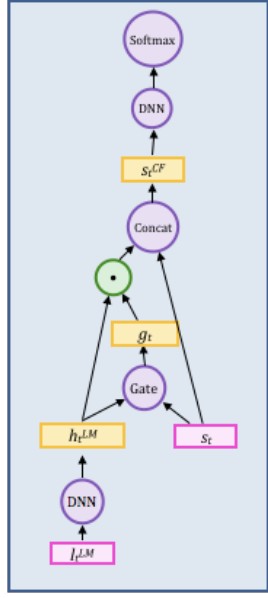

Figure 1: Cold Fusion Architecture

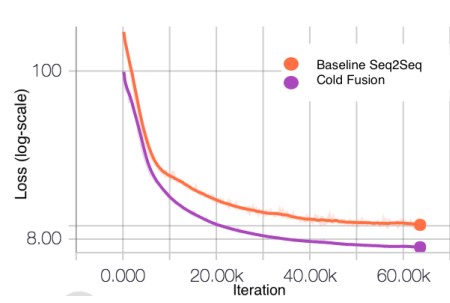

Figure 2: Cross-entropy loss on the dev set for the baseline model (orange) and the proposed model (purple) as a function of training iteration. Training with a language model speeds up convergence considerably.

Amazon Mechanical Turk to collect audio recordings of speakers reading out the text. We gave identical instructions to all the turkers in order to ensure that the two datasets only differed in the text domain. The source dataset contains 411,000 utterances (about 650 hours of audio), and the target dataset contains 345,000 utterances (about 676 hours of audio). We held out 2048 utterances from each domain for evaluation.

The text of the two datasets differ significantly. Table 2 shows results of training character-based recurrent neural network language models (Mikolov, 2012) on each of the datasets and evaluating on both datasets. Language models very easily overfit the training distribution, so models trained on one corpus will perform poorly on a different distribution. We see this effect in Table 2 that models optimized for the source domain have worse perplexity on the target distribution.

| Model | Domain | Word Count | Perplexity | |
|---|---|---|---|---|
| | | | Source | Target |
| GRU ($3 \times 512$) | Source | 5.73M | 2.670 | 4.463 |
| GRU ($3 \times 512$) | Target | 5.46M | 3.717 | 2.794 |
| GRU ($3 \times 1024$) | Full | 25.16M | **2.491** | **2.325** |

Table 2: Dev set perplexities for character RNN language models trained on different datasets on source and target domain. Note that i) the model trained on source domain does poorly on target domain and vice-versa indicating that the two domains are very different, and ii) the best model on both domains is a larger model trained on a superset of both corpuses. We use the model trained on the full dataset (which contains the source and target datasets along with some additional text) for all of the LM integration experiments.

## 4.2 NEURAL NETWORK ARCHITECTURES

The language model described in the final row of Table 2 was trained on about 25 million words. This model contains three layers of gated recurrent units (GRU) (Chung et al., 2014) with a hidden state dimension of 1024. The model was trained to minimize the cross-entropy of predicting the next character given the previous characters. We used the Adam optimizer (Kingma & Ba, 2014) with a batch size of 512. The model gets a perplexity of 2.49 on the source data and 2.325 on the target data.

| Model | Train Domain | Test on Source | | Test on Target | | |
|---|---|---|---|---|---|---|
| | | CER | WER | CER | WER | Domain Gap |
| Baseline CTC Model | | | | | | |
|   + Shallow Fusion | Source | 8.38% | 14.85% | 18.92% | 47.46% | |
| Baseline Attention Model | Source | 7.54% | 14.68% | 23.02% | 43.52% | 100% |
| Baseline Attention Model | Target | | | 8.84% | 17.61% | 0% |
| Baseline + Deep Fusion | Source | 7.64% | 13.92% | 22.14% | 37.45% | 76.57% |
|   + $s^{AM}$ in gate | Source | 7.61% | 13.92% | 21.07% | 37.9% | 78.31% |
|   + Fine-Grained Gating | Source | 7.47% | 13.61% | 20.29% | 36.69% | 73.64% |
|   + ReLU layer | Source | 7.50% | 13.54% | 21.18% | 38.00% | 78.70% |
| Baseline + Cold Fusion | | | | | | |
|   + $s^{AM}$ in gate | Source | 7.25% | 13.88% | 15.63% | 30.71% | 50.56% |
|   + Fine-Grained Gating | Source | 6.14% | 12.08% | 14.79% | 30.00% | 47.82% |
|   + ReLU layer | Source | 5.82% | **11.52%** | 14.89% | 30.15% | 48.40% |
|   + Probability Projection | Source | **5.94%** | 11.87% | **13.72%** | **27.50%** | **38.17%** |

Table 3: Speech recognition results for the various models discussed in the paper. The CTC model is based on Deep Speech 2 (Amodei et al., 2015) architecture.

| Model | Libris Test-Clean | | WSJ Eval-92 | | Target | |
|---|---|---|---|---|---|---|
| | CER | WER | CER | WER | CER | WER |
| Wav2Letter + Shallow Fusion | | | | | | |
|   MFCC | 6.9% | 7.2% | | | | |
|   Power Spectrum | 9.1% | 9.4% | | | | |
|   Raw Wave | 10.6% | 10.1% | | | | |
| Baseline Attention | 5.99% | 12.00% | 6.77% | 20.77% | 19.97% | 40.02% |
| Baseline + Deep Fusion | 5.75% | 11.31% | 6.23% | 18.82% | 18.62% | 37.21% |
| Baseline + Cold Fusion | **5.13%** | **9.92%** | **5.47%** | **16.68%** | **17.37%** | **34.20%** |

Table 4: Results from models trained on the publicly available Librispeech data. All of the acoustic models were trained on Librispeech training data and evaluated on librispeech test-clean, WSJ test-92 and our proprietary target domain data. Results from the Wav2Letter model (Collobert et al. (2016)) are presented for reference

For the acoustic models, we used the Seq2Seq architecture with soft attention based on Bahdanau et al. (2016). The encoder consists of 6 bidirectional LSTM (BLSTM) (Hochreiter & Schmidhuber, 1997) layers each with a dimension of 480. We also use max pooling layers with a stride of 2 along the time dimension after the first two BLSTM layers, and add residual connections (He et al., 2015) for each of the BLSTM layers to help speed up the training process. The decoder consisted of a single layer of 960 dimensional Gated Recurrent Unit (GRU) with a hybrid attention (Chorowski et al., 2015b).

The final Cold Fusion mechanism had one dense layer of 256 units followed by ReLU before softmax.

## 4.3 TRAINING

The input sequence consisted of 40 mel-scale filter bank features. We expanded the datasets with noise augmentation; a random background noise is added with a 40% probability at a uniform random SNR between 0 and 15 dB. We did not use any other form of regularization.

We trained the entire system end-to-end with Adam (Kingma & Ba, 2014) with a batch size of 64. The learning rates were tuned separately for each model using random search. To stabilize training early on, the training examples were sorted by increasing input sequence length in the first epoch

(Amodei et al., 2015). During inference, we used beam search with a fixed beam size of 128 for all of our experiments.

We also used scheduled sampling (Bengio et al., 2015) with a sampling rate of 0.2 which was kept fixed throughout training. Scheduled sampling helped reduce the effect of exposure bias due to the difference in the training and inference mechanisms.

## 4.4 IMPROVED GENERALIZATION

Leveraging a language model that has a better perplexity on the distribution of interest should directly mean an improved WER for the ASR task. In this section, we compare how the different fusion methods fare in achieving this effect.

Swapping the language model is not possible with Deep Fusion because of the state discrepancy issue motivated in Section 3. All fusion models were therefore trained and evaluated with the same language model that achieved a low perplexity on both the source and target domains (See Table 2). This way, we can measure improvements in transfer capability over Deep Fusion due to the training and architectural changes.

Table 3 compares the performance of Deep Fusion and Cold Fusion on the source and target held-out sets. Clearly, Cold Fusion consistently outperforms on both metrics on both domains than the baselines. For the task of predicting in-domain, the baseline model gets a word error of 14.68%, while our best model gets a relative improvement of more than 21% over that number. Even compared to the recently proposed Deep Fusion model (Gulcehre et al., 2015), the best Cold Fusion model gets a relative improvement of 15%.

We get even bigger improvements in out-of-domain results. The baseline attention model, when trained on the source domain but evaluated on the target domain gets, 43.5% WER. This is significantly worse than the 17.6% that we can get by training the same model on the target dataset. The goal of domain adaptation is to bridge the gap between these numbers. The final column in Table 3 shows the remaining gap as a fraction of the difference for each model.

The Deep Fusion models can only narrow the domain gap to 76.57% while Cold Fusion methods can reduce it to 38.17%. The same table also shows the incremental effects of the three architectural changes we have made to the Cold Fusion method. Note that applying the same changes to the Deep Fusion method does not yield much improvements, indicating the need for cold starting Seq2Seq training with language models. The use of probability projection instead of the language model state in the fusion layer substantially helps with generalization. Intuitively, the character probability space shares the same structure across different language models unlike the hidden state space.

We present additional results on the publicly available Librispeech dataset (Panayotov et al. (2015)) in Table 4. Librispeech contains 1000 hours of read audio speech and comes with train, development and test splits. We used similar architectures for the Seq2Seq model

Table 5: Effect of decoder dimension on the model's performance. The performance of cold fusion models degrades more slowly as the decoder size decreases. This corroborates the fact that the decoder only needs to learn the task not label generation. Its effective task capacity is much larger than without fusion.

| Model | Decoder size | Source | |
|---|---|---|---|
| | | CER | WER |
| Attention | 64 | 16.33% | 33.98% |
| | 128 | 11.14% | 24.35% |
| | 256 | 8.89% | 18.74% |
| | 960 | 7.54% | 14.68% |
| Cold Fusion | 64 | 9.47% | 17.42% |
| | 128 | 7.96% | 15.15% |
| | 256 | 6.71% | 13.19% |
| | 960 | 5.82% | 11.52% |

and the language model as in the previous experiments, but we trained the Seq2Seq model only on the 960 hour Librispeech training dataset.

Table 4 demonstrates that Cold Fusion can improve generalization compared to Deep Fusion on Librispeech data. We also demonstrate the superior domain adaptation capability of Cold Fusion by evaluating these models on the standard Wall Street Journal (WSJ) corpus and the target domain data described above.

Table 6: Results for fine tuning the acoustic model (final row from Table 3) on subsets of the target training data. *The final row represents an attention model that was trained on all of the target domain data.

| Model | Target Data | CER | Target WER | Domain Gap |
|---|---|---|---|---|
| Cold Fusion | 0% | 13.72% | 27.50% | 38.17% |
| Cold Fusion + finetuning | 0.6% | 11.98% | 23.13% | 21.30% |
| | 1.2% | 11.62% | 22.40% | 18.49% |
| | 2.4% | 10.79% | 21.05% | 13.28% |
| | 4.8% | 10.46% | 20.46% | 11.00% |
| | 9.5% | 10.11% | 19.68% | 7.99% |
| Attention* | 100% | 8.84% | 17.61% | 0.00 % |

## 4.5 DECODER EFFICIENCY

We test whether cold fusion does indeed relieve the decoder of learning a language model. We do so by checking how a decrease in the decoder capacity affected the error rates. As evidenced in Table 5, the performance of the Cold Fusion models degrades gradually as the decoder cell size is decreased whereas the performance of the attention models deteriorates abruptly beyond a point. It is remarkable that the Cold Fusion decoder still outperforms the full attentional decoder with $4\times$ fewer number of parameters.

Also, we find that training is accelerated by a factor of 3 (see Figure 2). Attention models typically need hundreds of thousands of iterations to converge (Chorowski et al., 2015a). Most of the training time is spent in learning the attention mechanism. One can observe this behavior by plotting the attention context over time and seeing that the diagonal alignment pattern emerges in later iterations. Because the pretrained, fixed language model infuses the model with lower level language features like the likely spelling of a word, error signals propagate more directly into the attention context.

## 4.6 FINE-TUNING FOR DOMAIN ADAPTATION

In the presence of limited data from the target distribution, fine tuning a model for domain transfer is often a promising approach. We test how much labeled data from the target distribution is required for Cold Fusion models to effectively close the domain adaptation gap.

The same language model from Section 4.4 trained on both the source and target domains was used for all fine-tuning experiments. The learning rate was restored to its initial value. Then, we fine-tuned only the fusion mechanism of the best Cold Fusion model from Table 3 on various amounts of the labeled target dataset.

Results are presented in Table 6. With just 0.6% of labeled data, the domain gap decreases from 38.2% to 21.3%. With less than 10% of the data, this gap is down to only 8%. Note that because we keep the Seq2Seq parameters fixed during the fine-tuning stage, all of the improvements from fine-tuning come from combining the acoustic and the language model better. It's possible that we can see bigger gains by fine-tuning all the parameters. We do not do this in our experiments because we are only interested in studying the effects of language model fusion in the Seq2Seq decoder.

Some examples are presented in Table 1. Recall that all models are trained on the source domain consisting of the read speech of search queries and evaluated on the read speech of movie scripts to measure out-of-domain performance. Because search queries tend to be sentence fragments, we see that the main mode of error for vanilla attention and Deep Fusion is due to weak grammar knowledge. Cold Fusion on the other hand demonstrates a better grasp of grammar and is able to complete sentences.

# 5 CONCLUSION

In this work, we presented a new general Seq2Seq model architecture where the decoder is trained together with a pre-trained language model. We study and identify architectural changes that are vital for the model to fully leverage information from the language model, and use this to generalize better; by leveraging the RNN language model, Cold Fusion reduces word error rates by up to 18% compared to Deep Fusion. Additionally, we show that Cold Fusion models can transfer more easily to new domains, and with only 10% of labeled data nearly fully transfer to the new domain.

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
