# OpenReview forum: "COLD FUSION: TRAINING SEQ2SEQ MODELS TOGETHER WITH LANGUAGE MODELS"
_ICLR.cc/2018/Conference — Invite to Workshop Track_

### Official Review · AnonReviewer1 · 2017-11-24

**Rating:** 5
**Confidence:** 5

**Review:**

The paper proposes a novel approach to integrate a language model (LM) to a seq2seq based speech recognition system (ASR). The LM is pretrained on separate data (presumably larger, potentially not the same exact distribution). It has a similar flavor as DeepFusion (DF), a previous work which also integrated an LM to a ASR in a similar way, but where the fusion is also trained. This paper argues this is not good as the ASR decoder and LM are trying to solve the same problem. Instead, ColdFusion first trains the LM, then fixes it and trains the ASR, so it can concentrate on what the LM doesn't do. This makes a lot of sense.

Experiments on private data show that the ColdFusion approach works better than the DeepFusion approach. Sadly these experiments are done on private data and it is thus hard to compare with benchmark models and datasets.

For instance, it is possible that the relative capacity (number of layers, number of cells, etc) for each of the blocs need to vary differently between the baseline, the ColdFusion approach and the DeepFusion approach. It is hard to say with results on private data only, as it cannot be compared with strong baselines available in the literature.

Unless a second series of experiments on known benchmarks is provided, I cannot propose this paper for acceptance.

***********
I have read the revised version. I applaud the use of a public dataset to
demonstrate some of the results of the new algorithm, and for this I am raising
my score. I am concerned, though, that while ColdFusion is indeed better than
DeepFusion on LibriSpeech, both of them are significantly worse than the
results provided by Wav2Letter on word error rates (although better on
character error rates, which are usually not as important in that literature).
Is there any reason for this?

---

### Official Review · AnonReviewer2 · 2017-11-26
**Better integration of language models into sequence 2 sequence networks.**

**Rating:** 6
**Confidence:** 5

**Review:**

The paper proposes a new way of integrating a language model into a seq2seq network: instead of adding the language model only during decoding, the model has access to a pretrained language model during training too. This makes the training and testing conditions more similar. Moreover, only the logits of the pretrained language model are used, making it possible to swap language models post-training.

The experiments show that the proposed language model fusion is effective, and works well even when different, domain-dependent language models are used during training and testing. Further experiments indicate that through the integration of a language model at training time the seq2seq's decoder can be smaller as it is relieved of language modeling.

Quality:
The paper is well executed, the experiments do basic validation of the model (ablation plus a specially designed task to show model effectiveness)

Clarity:
Well written, easy to understand.

Originality:
The main idea is new.

Significance:
Better language model integration and easier adaptation to new domains of seq2seq models is important.

Pros and cons:
pros : see above

cons:
My problem with the paper is lack of experiments on public datasets. The efficacy of the method is shown on only one task on a proprietary corpus engineered for domain mismatch and the method may be not so efficient under other circumstances.  Besides presenting results on publicly available data, the paper would also be improved by adding a baseline in which the logits of the language model are added to the logits of the seq2seq decoder at training time. Similarly to cold-fusion, this baseline also allows swapping of language models at test time. In contrast, the baselines presented in the paper are weaker because they don't use a language model during training time.

---

### Official Review · AnonReviewer3 · 2017-11-27

**Rating:** 5
**Confidence:** 5

**Review:**

This paper present a simple but effective approach to utilize language model information in a seq2seq framework. The experimental results show improvement for both baseline and adaptation scenarios.

Pros:
The approach is adapted from deep fusion but the results are promising, especially for the off-domain setup. The analysis also well-motivated about why cold-fusion outperform deep-fusion.

Cons:
(1) I have some question about the baseline. Why the decoder is single layer but for LM it is 2 layer? I suspect the LM may add something to it.  For my own Seq2seq model, 2 layer decoder always better than one. Also, what is HMM/DNN/CTC baseline ? Since they use a internal dataset, it's hard to know how was the seq2seq numbers. The author also didn't compare with re-scoring method.

(2) It would be more interesting to test it on more standard speech corpus, for example, SWB (conversational based) and librispeech (reading task). Then it's easier to reproduce and measure the quality of the model.

(3) This paper only report results on speech recognition. It would be more interesting to test it on more area, e.g. Machine Translation.

Missing citation: In (https://arxiv.org/pdf/1706.02737.pdf) section 3.3, they also pre-trained RNN-LM on more standard speech corpus. Also, need to compare with this type of shallow fusion.

Updates:

https://arxiv.org/pdf/1712.01769.pdf (Google's End2End system) use 2-layer LSTM decoder.
https://arxiv.org/abs/1612.02695,  https://arxiv.org/abs/1707.07413 and https://arxiv.org/abs/1506.07503) are small task.
Battenberg et al. paper (https://arxiv.org/abs/1707.07413) use Seq2Seq as a baseline and didn't show any combined results of different #decoder layer vs. different LM integration method. My point is how a stronger decoder affect the results with different LM integration methods. In the paper, it still only compared with deep fusion with one decoder layer.

Also, why it only compared shallow fusion w/ CTC model? I suspect deep decoder + shallow fusion already could provide good results. Or the gain is additive?

Thanks a lot adding Librispeech results. But why use Wav2Letter paper (instead of refer to a peer reviewed paper)? The Wav2letter paper didn't compare with any baseline on librispeech (probably because librispeech isn't a common dataset, but at least the Kaldi baseline is there).

In short, I'm still think this is a good paper but still slightly below the acceptance threshold.

---

### Decision · Program_Chairs · 2018-01-29
**ICLR 2018 Conference Acceptance Decision**

**Decision:**

Invite to Workshop Track

**Comment:**

Pros
-- A novel way to incorporate LM into an end-to-end model, with good adaptation results.

Cons
-- Lacks results on public corpora or results are not close to SOTA (e.g., for Librispeech).

Given the reviews, it is clear that the experimental evaluations can be improved. But the presented approach is novel and interesting. Therefore I am recommending the paper to the workshop track.